# Self-Assembly, Self-Folding, and Origami: Comparative Design Principles

**DOI:** 10.3390/biomimetics8010012

**Published:** 2022-12-27

**Authors:** John R. Jungck, Stephen Brittain, Donald Plante, James Flynn

**Affiliations:** 1Department of Biological Sciences, University of Delaware, Newark, DE 19716, USA; 2Department of Mathematical Sciences, University of Delaware, Newark, DE 19716, USA; 3Department of Mathematics, University of New Hampshire at Manchester, Manchester, NH 03101, USA

**Keywords:** self-assembly, self-folding, origami, 4D printing, polyhedra, topology, Dürer nets, Schlegel diagrams, Hamiltonian circuits, Eulerian paths

## Abstract

Self-assembly is usually considered a parallel process while self-folding and origami are usually considered to be serial processes. We believe that these distinctions do not hold in actual experiments. Based upon our experience with 4D printing, we have developed three additional hybrid classes: (1) templated-assisted (tethered) self-assembly: e.g., when RNA is bound to viral capsomeres, the subunits are constricted in their interactions to have aspects of self-folding as well; (2) self-folding can depend upon interactions with the environment; for example, a protein synthesized on a ribosome will fold as soon as peptides enter the intracellular environment in a serial process whereas if denatured complete proteins are put into solution, parallel folding can occur simultaneously; and, (3) in turbulent environments, chaotic conditions continuously alternate processes. We have examined the 43,380 Dürer nets of dodecahedra and 43,380 Dürer nets of icosahedra and their corresponding duals: Schlegel diagrams. In order to better understand models of self-assembly of viral capsids, we have used both geometric (radius of gyration, convex hulls, angles) and topological (vertex connections, leaves, spanning trees, cutting trees, and degree distributions) perspectives to develop design principles for 4D printing experiments. Which configurations fold most rapidly? Which configurations lead to complete polyhedra most of the time? By using Hamiltonian circuits of the vertices of Dürer nets and Eulerian paths of cutting trees of polyhedra unto Schlegel diagrams, we have been able to develop a systematic sampling procedure to explore the 86,760 configurations, models of a T1 viral capsid with 60 subunits and to test alternatives with 4D printing experiments, use of Magforms^TM^, and origami models to demonstrate via movies the five processes described above.

## 1. Introduction

Self-assembly is an important biological phenomenon involved in the formation of viral capsids, ribosomes, mitotic spindles composed of microtubules, molecular motors and other associated proteins that segregate chromosomes, and membranes. Self-folding is involved in DNA, RNA, and protein folding as well as in macroscopic activities such as wing-unfolding in beetles, leaf unfolding, seed release in pinecone drying, poppy petal blossoming, and closing of Venus flytraps. Questions about self-folding and self-assembly, while simple to pose, can be more complex than originally supposed. Furthermore, the processes of self-assembly and self-folding can often be inter-related. Herein we define self-assembly to be the assembly of complex structures from elementary building blocks without significant external intervention and we define self-folding as a branch of self-assembly that is constrained by bending and binding at specific points within building blocks. We investigated distinctive differences between self-assembly, self-folding, and origami to develop a better understanding of principles of these types of self-organization, and to better design structures that have potential applications in bioengineering research and biomedical applications such as drug delivery, additive manufacturing fabrication strategies, and compact emergency shelters that can be delivered and easily constructed in relief and refugee centers in remote locations or extra-terrestrially.

A difficult educational concept and a historical scientific problem is: “How can complex organized structures form spontaneously?” Counterintuitively, self-assembly and self-folding of complex biological structures is entropy driven. John Pelesko [1] notes that “nature repeatedly uses the same motif in designing systems [namely:] … structured particles, a binding force, an environment, and a driving force. Self-assembly and self-folding have also emerged as fundamental processes in the current engineering approach to advanced manufacturing processes: “self-assembly is one of the few practical strategies for making ensembles of nanostructures” [2]. For mathematical students, self-assembly fits generically as a bottom-up form of modeling.

We have been stimulated to develop a better understanding of fundamental principles of self-assembling and self-folding polyhedra because both processes are involved in forming the capsids of viruses. The work of Caspar and Klug ([3] classified icosahedral viral capsids according to their triangulation number, while the work of Bonnie Berger et al. [4] argued that perhaps there are local rules that govern the assembly of virus shells globally. Twarock et al. [5] extended Caspar and Klug’s work to deal with quasicrystal viral capsids and RNA-tethered viral capsomeres that combine aspects of self-assembly and self-folding. This process is more general and has been called “template-assisted self-assembly” [6,7,8]. Another description of supramolecular self-assembly has been identified as “hierarchical self-assembly” by Sun et al. [9]. Another distinctive process is associated with proteinaceous fiber structures such as collagen, amyloid, and actin filaments and has been called “unidimensional assembly” [10]. Herein we assert that further classifications of self-assembling and self-folding yield insight into the behavior of such processes because there is a need to understand that these descriptions of supramolecular formation of configurations are not mutually exclusive.

In order to distinguish between multiple aspects of designing and utilizing self-organizing models, we illustrate herein polyhedral models that we have designed which feature self-assembly, self-folding, and hybrids between them in order to develop both geometric and topological principles of these models of self-organization.

When Skylar Tibbits, the founder of the Self-Assembly Lab at MIT, published his book: Things Fall Together: A Guide to the New Materials Revolution [11], it was heralded as “a manifesto for the dawning age of active materials”. Reviewer Kevin Kelly argues that Tibbits “demonstrates that the seemingly wild idea of a biology-like technology is not impossible”. Tibbits [12] distinguished between self-folding and self-assembling processes as well as human folded origami thusly:

“Material parts that are designed in pre-connected configurations such as strands, fibers, sheet materials, and cast objects that can be designed to change shape and appearance through mechanical joints. … [that] allow folding, curling, shrinking, expanding, and other active material transformations.

Autonomous components that are moving around and need to find one another, connect or disconnect and then error-correct. … This is the logical antithesis to human assembly that requires skilled placement and directed energy to go from arbitrary components into a final form. Self-assembly allows materials to spontaneously assemble without pick-and-place guidance”. [12] (p. 75).

We argue that Tibbits’ [12] classification should be extended to include hybrid processes. Thus, we present Table 1 (below) of the different types of assembling and folding, the type of assembly pathway they represent, whether or not they are guided by an external force (such as an external force like scaffolding proteins), how they have been studied in the literature and in the lab, and whether or not there are any known biological analogues to a particular type of folding.

In this paper we focus on the construction of physical models for parallel self-assembly and serial self-folding and their design principles. In the future we plan to construct models that are also hybrid in nature, such as those that self-fold into subunits and then self-assemble into complete designs as well as template assisted serial folding.

## 2. Serial Folding

Human or guided serial folding is a folding process guided by some intelligent design or external mechanism. In this method of folding, a guiding force directs the folding of the net along a specific folding pathway where only one or very few folds occur simultaneously. As people have only two hands, anyone physically folding a net or doing origami is an example of guided serial folding, since they can only fold as many faces as they can control in their hands, and therefore must proceed along the net in some logical manner. Of course, the guiding force does not have to be human. A robotic arm(s) trained to fold polyhedron nets would be considered the same type of folding.

In traditional origami models of polyhedral viral capsids, the polyhedron begins as a flat sheet in the configuration of a Dürer net (Figure 1). A Dürer net has four properties: “(1) The net is planar. (2) It is a single piece. (3). It is the result of cutting polyhedron edges. This is called an edge unfolding. (4) It is non-self-overlapping in the plane, so that uncut points do not unfold on top of one another”. (O’Rourke, [13] (p. 106)) As of that publication, it was still an open problem of whether every convex polyhedron has an edge unfolding to make a planar net. A variety of such origami models of various polyhedral viral capsids is available for the Protein Data Base (PDB) site: https://pdb101.rcsb.org/learn/paper-models/quasisymmetry-in-icosahedral-viruses, [accessed on 1 December 2022] on that uses the Caspar and Klug [3] geometric approaches of laying out the individual capsomeres on a hexagonal graph (Figure 2).

For models of the unit cell of 15 different polyhedral viral capsids see the Viral Zone website (Figure 3).

Dürer nets are planar projections of polyhedra that preserve the geometrical properties: lengths, areas, and angles. The number of configurations explodes combinatorially (Table 2).

In order to understand which polygonal face binds to another, we use the relationship between the Schlegel diagram and the topology of the corresponding Dürer net. In Figure 4, we have numbered each pentagon of a dodecahedron on the corresponding Schlegel diagram and the corresponding Dürer net to be the same. These maps help guide the folder in converting the planar origami model into its 3D polyhedron.

The basic question for an origami-ist is: which Dürer net is easiest to fold in the shortest period of time? Since only two edges can be attached at a time, folding is a serial process. For a human folder, three different challenges arise: (i) how many edges need to be glued?; (ii) how easy is it to successively attach edges to finish building a complete polyhedron? and, (iii) which configurations produce intermediate 3D structures that if not glued in a particular order prevent the construction of a complete polyhedron? Consider two different Dürer nets in Figure 5.

Before leaving human-mediated folding, we have been informed by the research on using Rubik’s snake to study protein folding [14] and subsequent research in robotics on reconfigurable order [15,16,17,18]. For example, Li, Hu, and Bishop [17] show that in order to convert the 1D sequence of Rubik’s snake into a 3D globular model of a protein, the 3D “ball” structure is generated by a series of rotational folds = [1, 3, 3, 1, 3, 1, 1, 3, 1, 3, 3, 1, 3, 1, 1, 3, 1, 3, 3, 1, 3, 1, 1, (3)], where “the last rotation in this sequence is marked in parentheses. This indicates that the “ball” is a closed loop. Closure requires the first and last wedges to be pinned”. Note that the linear configuration of the Dürer net of an icosahedron in Figure 4a is not so dissimilar from 1D configuration of Rubik’s snake. Li, Hu, and Bishop [17] conclude: “The obvious next steps are to develop methods for collision detection, to explore the effects of different wedge geometries, and to consider nonuniform embeddings. The methods presented here are sufficient to demonstrate a systematic approach for achieving multidimensional (1D and 3D) multi-scale modeling (via embedding) of the chain-like slender body objects”. These same issues of: (i) collision detection; (ii) different geometries; and, (iii) nonuniform embeddings have arisen in our research and are addressed below.

## 3. Radial Serial Folding

Another way to model origami that utilizes self-folding is to use magnetic polygons available from a number of different toy companies. It is easy to build 3D polyhedral from different 2D polygons (tetrahedra, octahedron, icosahedra from triangular pieces, cubes from squares, dodecahedra from pentagons, etc.). These models are often used in education to help students learn Euler’s formula which applies to all convex polyhedra. We have found that we could model the process of “Radial Serial Folding” with such models (Figure 6). These high school experiments show that you can pick up the net from a single polygon and the whole polyhedron comes together without any other necessary input. In this example, gravity drives the folding of the innermost edges first, with the outer panels eventually lifting off the table and swinging in to complete the finished shape. Assembly pathways where folding begins towards the center of mass of the net and travels outward is what we consider to be radial serial folding.

Due to the limitations of inertia on nets parallel folding nets in solution, we mentioned that parallel folding in experiment often resembles inverted radial folding, where folding starts at the edges of the net and travels inward towards the center of mass.

By using Magformers^TM^ [19] to construct many different Dürer nets of polyhedra, we could experiment and find examples such as in Figure 6 that when lifted easily assembled into full polyhedrons while almost all others fell apart upon lifting one polygon or only partially folded. On the other hand, we use Magformers^TM^ [19] to illustrate that simply having pieces that can form a structure does not mean that they are analogous to self-assembling models. For example, in Figure 7, we show how that if we mix equilateral Magformer^TM^ units in a vessel, that we are more apt to generate lots of tetrahedra, octahedra, and decahedra, with only a few incomplete icosahedra and no T1 60-piece icosahedra.

Since Magformers^TM^ [19] are quite heavy, every long Dürer net of either dodecahedra or icosahedra that we tried was never able to fold by itself into a complete structure. However, these experiments informed our 4D printing experiments in choosing which Dürer net configurations to initially try.

## 4. True Parallel Self-Assembly and Self-Folding

### 4.1. Parallel Self-Assembly

Palma, Cecchini, and Samorì [20] assert that: “Self-assembly is one of the most important concepts of the 21st century”. We [21] have asserted that the entropy-driven self-assembly of many biological structures counters traditional narratives that assert biological organization is due to working against the second law of thermodynamics and helps us understand the evolution of many complex biological structures. Thus, models of self-assembly are important in helping scientists and students [22] better understand how complex biological patterns (“designs”) can result from random interactions. Furthermore, as noted by Swiegers, Balakrishnan, and, Huang [23]:
“thermodynamic self-assembly … involves the establishment of a kinetically rapid, reversible, thermodynamic equilibrium… which results in the energetically most stable product being formed in the greatest proportions. Because the equilibrium is reversible, the individual coordinate bonds need not form in the desired manner each and every time. Instead, the constant forming and reforming of bonds … results in ‘incorrect’ bonds being undone and associating ‘correctly’ under a thermodynamic impetus. Thermodynamic self-assembly therefore has the unique property of being ‘self-correcting.’ … the key to using this class of self-assembly as a synthetic tool is to ensure that the desired product will be more stable than any possible competing product. … the [more that] the desired product is selectively favored, the greater its stability relative to its competitors, the greater its proportion in solution”.

Protein subunits (capsomeres) of polyhedral viral capsids vary in their configuration, so we decided to focus on the 20 simple equilateral triangular pieces of an icosahedron. If you look at the detailed protein structure of capsomeres of a T1 virus such as the Satellite Tobacco Necrosis Virus (Figure 8), the inter-actions between capsomeres involve non-covalent boding through ionic interactions, positive-negative polar associations, hydrogen bonding, and van der Waals hydrophobic residue interactions. Thus, we used magnets between our capsomere models as with Olson’s group’s interactive meso-scale models of self-assembling polyhedral viral capsids [22,24,25,26] at the Scripps Research Institute in La Jolla, CA. They produced beautiful self-assembling dodecahedra. As delightful as these models have been for both the research and education they stimulated, we felt that icosahedral models would better represent the different members of the polyhedral viral capsid family. Thus, as original work in this area, we have modeled the self-assembly of a T1 Satellite Tobacco Necrosis Virus in three different ways.

First, we [27] constructed a self-assembling icosahedron (Figure 9; Appendix A). We use a Schlegel diagram (Figure 10) to construct a topological model of where to place the magnets so that all identical subunits align and bind. As was previously done with Olson’s dodecahedron model [22,24,25,26], by reversing the location of north and south poles of all of our magnets, we can construct two different enantiomers of the self-assembling icosahedron. See Tibbits [11] (pp. 80–82) for a similar experiment with two different enantiomers of the self-assembling dodecahedron.

After numerous unsuccessful attempts to produce self-assembling polyhedral, we have developed five design criteria:

First criterion: we must consider the need for a magnet map of complementary attracting and repulsing orientations. To determine which magnets should attract each other to form our complete structure, we used Schlegel diagrams because they preserve the topology of the configuration so that we can determine which subunits specifically bind to one another.

Second criterion: maintain a rotational symmetry of each subunit in the model.

Third criterion: design each subunit so that they are all identical to each other.

Fourth criterion: design each subunit so that it assembles into a sphere. We have found that having subunits with convex faces increases the odds that the magnets in two separate subunits will interact with each other when shaken in a container.

Fifth criterion: Self-assembly occurs in an environment. Therefore, the configuration of the vessel (both its shape and size) that the pieces are shaken in matters enormously in determining the time to produce full polyhedral. This is akin to a “ship-in-a-bottle” problem. (Table 3 and Figure 11).

Thus, self-assembly occurs not only much faster on average in spherical containers than in cylindrical containers, but the several of the experiments in cylindrical containers were not successful in generating complete icosahedra.

Combined, these five criteria help to increase the speed at which a given design correctly self-assembles, as the pieces can assemble in many different orientations. For each of the five Platonic solids one can glue a north facing and a south facing magnet on each edge of a convex subunit to achieve these goals. We have, for the first time, successfully produced self-assembling tetrahedra, cubes, octahedra, dodecahedra, and icosahedra models based upon these criteria.

However, to better model polyhedral viral capsids we wanted to build models using rhombic cells (icosahedral asymmetric units) such as shown in Figure 3 for a T1 virus. Therefore, we built a self-assembling decahedron of ten rhombi made by simply dividing a sphere into the 20 equilateral triangular faces but keeping ten pairs glued together (Figure 12a) and by developing asymmetric rhombi similar to those illustrated in many virology journals (Figure 12b).

The decahedron model of subunits (Figure 12c) with knobs and holes was an important intermediate step in our research to develop a full sixty subunit T1 icosahedral model (Figure 13).

Self-assembly in each of these cases is a parallel process as the subunits are simply shaken together in a container with no one directing which subunit is interacting with any other one. To avoid human mediation, we often do the shaking in a rock tumbler or shaker bath to maintain randomness. When we do shake subunits by hand and intermediate dead-ends occur (such as four pentagonal assemblies of five triangular pieces), in order to assemble a full icosahedron, we break them and start over again. We also did this when using the rock tumbler because it becomes apparent after a certain amount of time tum-bling that some “bad” intermediates will not break apart on their own. We believe that these biomimetic models of self-assembling polyhedral not only help us better understand how viral capsids assemble in vivo, but illuminate how we might build nano-polyhedra for drug delivery that both assemble a priori and easily disassemble a posteriori at a target site with an external stimulus.

### 4.2. Parallel Self-Folding

We tackled the self-folding portion of this problem, thinking about self-folding nets of the platonic solids in an attempt to clarify and consolidate the existing literature, standardize existing notation, and provide new insights on what nets fold best.

By 3D printing a Dürer net of an icosahedron from polypropylene filament in two successive layers (the first layer is thinner and be-comes a series of hinges) (Figure 14) and dropping it into warm water, the net automatically folds into a 3D icosahedron. For such self-folding models we have had to set the direction of the first printed layer so that it is not parallel to any edge in the net. If the first (hinge) layer is printed with lines parallel to the direction of the living hinge, then the hinge will quickly break.

Prior to our work, many of the self-folding models in the literature required high tech spaces involving tiny models made from difficult to use materials such as solder or DNA. Our models can be printed in a middle school with a $50 roll of polypropylene filament and $10 of magnets. This work makes self-folding models much more accessible.

To consider self-folding, we need to understand that the number of Dürer nets explodes combinatorially (Table 2 above). Thus, the primary design challenge is to determine which Dürer net is able to fold faster to form a complete polyhedron with the least chance of forming intermediates that are unable to continue to self-fold. To distinguish each of the 43,380 Dürer nets of dodecahedra and the 43,380 Dürer nets of icosahedra, we use the Hamiltonian circuit of the degrees of the vertices (Figure 15).

Unfortunately, there is a confusion in the literature in the definition of a vertex connection (Figure 16). Many articles define a vertex connection as a place on the net where two faces share a vertex but do not share an edge [29,30]. This works well for nets of the dodecahedron where there is only one way for such a joining of faces to occur, creating a 36° angle. This is also fine for nets of a cube, in which vertex connections only occur with the formation of 90° angles. Applying this same definition to nets with triangular faces results in two different types of vertex connections for the octahedron and three different types of vertex connections for the icosahedron. In the case of the octahedron, under the classical definition, vertex connections can be formed when three faces share a vertex forming a 180° angle, and when four faces share a vertex forming a 120° angle. For our purposes, we wish to only consider the more acute angle. Likewise, the first definition of vertex connections allows 180°, 120°, and 60° angles to be considered vertex connections for the icosahedron. Categorizing all three of these different types of connections as vertex connections sacrifices specificity of categorization that we do not wish to allow. Specifically, we desire a second, generalized definition for vertex connections that results in only one type of angle vertex connection and can be generalized not only to the platonic solids but to the Archimedean solids as well. So for the dodecahedron we consider only 36° degree angles, formed at vertices of degree 4 in the Durer net, and for the icosahedron 60° degree angles, formed at vertices of degree 6 in the net. From this point forward the term vertex connection of a net is used to refer to a vertex of a net that is incident in that net to every face it will be incident to in the assembled polyhedron. That is, at a vertex connection, the only operation needed to be performed to complete the polyhedron locally in a ϵ ball about that vertex is the gluing of two edges to each other. The number of vertex connections of a net is then the number of vertices of degree 3 for a cube net, degree 5 for a cube net, degree 4 for a dodecahedron net, and degree 6 for an icosahedron net.

We can also measure the number of leaves on the spanning tree of a net. This is also a variable that highly relates to the compactness of the net and how well the net folds. A leaf only shares one edge with other polygons in a Dürer net.

All eleven Dürer nets of cubes and octahedra have been extensively studied by Dodd, Damasceno, and Glotzer [29]. They focus on the number vertex connections on each net (Figure 17).

Spanning trees are a concept from graph theory that can be applied to our self-folding situation as a way to represent Dürer nets. A graph G = {V, E} is a collection of vertices V and a collection of edges E indicating which vertices have an edge between them. A spanning tree T = {V′, E′} of a graph G is a subgraph of G (That is V′ ⊆ V and E′ ⊆ E) such that V′ = V, every vertex has at least one incident edge, and T has no cycles. That is, we say T is a spanning tree if it contains all the vertices of the original graph, and every vertex still has at least one edge connected to it, but there are paths that can return to their starting vertex without repeating a vertex along the way.

There are two different types of spanning trees we want to consider. The first type of spanning tree is that of a cutting tree. A cutting tree represents the edges you would cut along to unfold a solid into a planar Dürer net. If we created any cycles in this tree, then we would completely detach a face or a collection of faces from the rest of the net, so certainly our cutting tree cannot contain any cycles. Additionally, the Dürer net must lie planar across its entirety. Therefore, since any closed solid is not locally planar at each vertex, at least one edge must be cut along so that the net will lie planar there. Therefore, every vertex of the polyhedron must be visited with no cycles. So the cutting tree is a spanning tree of the graph G = {V, E} where V is the set of the polyhedron’s vertices and E is the set of the polyhedron’s edges.

Another way to visualize the relationship of a spanning tree on a 3D polyhedron to a 2D Dürer net is that instead of doing an edge cut, we can cut 3 of the 6 faces of a cube to form 6 triangular faces. The resulting Dürer net has a spanning tree of length 9 instead of 4 to 6 as shown in Figure 17 (Figure 18).

The other type of spanning tree is a skeleton graph. In this case, we take the Dürer net itself and create a graph that represents it in the plane. We place one vertex at the center of each face on the Dürer net, and we connect two vertices with an edge if those two faces are connected in the planar Dürer net (Figure 1, Figure 2, Figure 14, Figure 15, Figure 16, Figure 17 and Figure 19). Since the net must lie planar and every face must be included for it fold to the completed polyhedron, the skeleton graph is a spanning tree.

A spanning tree helps us generate two other useful topological invariants of Dürer nets. In graph theoretic terms, the diameter of a graph is the maximum taken over all pairs of vertices $\left(v_{i},v_{j}\right)$ of the minimum distance between $v_{i}$ and $v_{j}$. However, since the skeleton graph is a spanning tree, there are no cycles and therefore it sufficient to define the diameter as just the length (measured in edges) of the longest path possible that does not repeat any vertices. We could consider this path the backbone of the skeleton (Figure 19).

For a given tree, it is possible for there to be multiple maximal paths. Thus, some skeleton graphs can have multiple backbones.

Another useful property of a spanning tree is its Degree Distribution. If we consider the spanning tree, in graph theoretic terms, then we can consider what the possible degrees of the vertices of the spanning tree of a Dürer net could be. A cube has 6 faces ⇒ Spanning tree has 5 edges
$$\underset{v\in V}{\sum }d\left(v\right)=2e=10$$

If A is the number of vertices of degree 1 (leaves), B = number of vertices of degree 2, C = number of vertices of degree 3, etcetera, then we can therefore write two linear equations:A + 2B + 3C + 4D = 10
A + B + C + D = 6

This system of equations has only 4 discrete solutions that do not violate linear algebra and that can be represented as a valid net. In Table 4, we show the four satisfactory solutions and in Table 5 we give the number of possible degree distributions for each type of Platonic solid.

Menon et al. [31], Pandey et al. [32], Kaplan et al. [33], and Pigrim et al. [34] used vertex connections and geometrically, the radius of gyration (Figure 20). Pandey et al. [32] define a net Ω’s radius of gyration by the following integral which computes a measure of how well packed the net is around its center of mass:$$R_{g}^{2}=\int_{\Omega }{\left(x-\bar{x}\right)}^{2}+{\left(y-\bar{y}\right)}^{2}dA$$

Other researchers [35] have slightly simplified this formula, by taking a summation over the centroids of the faces of the net instead of an integral over the whole net. We use this computation instead of the integral one because it produces nearly the same ranking of nets from least to most compact, while only sacrificing a small amount of specificity.
$$R_{g}^{2}=\frac{1}{N}\overset{N}{\underset{i=1}{\sum }}\left[{\left(x_{i}-\bar{x}\right)}^{2}+{\left(y_{i}-\bar{y}\right)}^{2}\right]$$

In this formula, N is the number of faces in the Dürer net, $\left(x_{i},y_{i}\right)$ is the center of the ith face of the Dürer net under some arbitrary enumeration of the faces, and $\left(\bar{x},\bar{y}\right)$ is the center of mass of the entire net. This formula does the same computation of the first, but instead of integrating over the entire net, it just performs the calculation for the center of each face. Regardless of definition used in the literature, several independent studies have found a strong correlation between a decrease of $R_{g}$ and an increase in the rate of successful foldings. In our data on the radius of gyration for various Dürer nets, we used the computationally simpler summation equation. One discrepancy in the literature is that both of these formulas are quadratically dependent on the side length of the faces of the net. Therefore, when researchers calculate the radius of gyration for a net, it is comparable to other nets in their paper, but may not be comparable to another researcher’s results who has used a different side length in their experiment. Therefore, we propose a normalization of the side lengths to 1 unit in this calculation. This allows radius of gyration of nets to be comparable between experiments even if the nets are printed physically with different side lengths. Regardless of definition, several independent studies have found a strong correlation between a decrease of $R_{g}$ and an increase in the rate of successful foldings. Another way we can measure compactness geometrically is by considering the convex hull of a Dürer net. The convex hull of a Dürer net is the smallest convex polygon that encloses it (Figure 21).

We can measure both the area and the perimeter of the convex hull. Since the side lengths of a Dürer net are fixed to 1 unit in all of our calculations, then the area of the convex hull measures how much empty space is added that is not part of the Dürer net. This gives us a measure of how much space is “wasted” and thus a measure of compactness. However, the smallest area of a simple polygon is typically a thin rectangular shape, so while radius of gyration prefers more spherical compactness, the area of the convex hull picks up on more 1-dimensional compactness.

On the other hand, perimeter is minimized the closer the convex hull comes to being circular. Therefore, the perimeter of the convex hull has a very high correspondence with the radius of gyration. Graphs of linear regression of area and perimeter of convex hulls to radius of gyration.

Therefore, measuring radius of gyration, area of the convex hull, and the perimeter of the convex hull all tell us different things about how we might consider a particular Dürer net to be compact.

These geometric factors, however, are often driven by topological ones, which are the main factors we consider herein. These topological factors are the number of vertex connections and number of leaves on the net, as well as the degree distribution (Table 4) and diameter of the spanning tree of the net (Figure 16). Instead of using the radius of gyration as a geometric factor, we use the more visually easier to conceptualize: the area and perimeter of a convex hull of the net. Statistically, both the area and perimeter of a convex hull of the net are so positively correlated with their respective radius of gyration that we do not feel anything is lost (Figure 22 and Figure 23).

This confirms our hypothesis that the radius of gyration and area of the convex hull measure two different types of geometric compactness [35], whereas the perimeter of the convex hull captures an extraordinarily similar type of compactness as the radius of gyration. Since the relationship between the perimeters of their respective convex hulls was much stronger than that of areas and captured much of the sense of the radii of gyration, we primarily use the perimeter of the convex hull as our geometric measure in our exploration of folded structures.

As defined above, we have identified seven topological and geometric variables that play a role in measuring a nets compactness and how well it folds.
vertex connectionsleaveslength of spanning treedegree distributionarea of a convex hullperimeter of a convex hullradius of gyration

While Menon et al. [31], Pandy et al. [32], Kaplan et al. [33], and Pigrim et al. [34] analyzed the folding of dodecahedra and icosahedra, they only examined models with the maximal number of vertex connections. Therefore, we chose to investigate the examples of self-folding of dodecahedra of all nine categories of Dürer nets from 2 to 10 vertex connections with different perimeters of their convex.

In experiment, true parallel self-folding rarely occurs due to limitations induced by inertia, and thus more closely resembles inverted radial serial folding. In Figure 24, even if folding begins simultaneously across the surface of the net, the outer panels will begin to lift first. This is because for the net to fold along the red-blue edge, only one panel must be lifted. However, to fold along the red-green edge, 2 panels must lift, and to fold against the green-yellow edge the assembly pathway has to resist the inertia of three panels. Ultimately, to form the folded polyhedron, the outer panels need to travel farther than the inner panels.

In addition to the 3D-printed Dürer net of an icosahedron that we described above (Figure 14) that successfully self-folds into a complete icosahedron, we have successfully printed two different Dürer nets of dodecahedra that partially and fully self-fold (Figure 25).

## 5. Random Folding

Random folding occurs when there is no one ideal assembly pathway, or the assembly pathway is determined by random events in the environment around the net. For example, a net that folds while subject to natural or simulated turbulence (Löthman et al., [36]) where panels are pushed and folded not by intelligent design but by random forces.

## 6. Conclusions

Kuang et al. [37] assert that: “theoretical models and design methodology are needed to accurately predict and optimize the shape shifting … Biomimetic design for 3D printing, as powerful tools for building hypothetical models, can not only facilitate better understanding of the biological function in nature but also inspire the creation of high-performance materials and functional engineering materials”. We believe that our explorations add significantly to the literature by moving beyond self-assembly of cubes and dodecahedra and beyond self-folding of cubes, octahedra, and a few dodecahedra and icosahedra.

We are extending our work in three directions: (1) More extensive, and conscious sampling from the 43,380 Dürer nets of dodecahedra and icosahedra. For example, we are examining 27 dodecahedra with three examples each from 2 to 9 vertex connections with quite different perimeters of their convex hulls. (2) Analysis of the thermodynamics and kinetics of self-assembly and self-folding. Besides our movies of self-assembly in different shapes and volumes of containers and experiments with shakers and rock-tumblers, we plan to build a device similar to Lothman et al.’s [36] and Abelmann et al.’s [38] self-assembly reactor so that we use video data to explore the impact of various degrees of turbulence. They conclude that their work “implies that we can generalize the outcome of these experiments to the design of electrostatically interacting objects of micrometer size for 3D self-assembly, aimed at applications such as photonic crystals, supermaterials, 3D electronics, or memories”. Re extending our self-folding experiments, we will not only vary the temperature of the water that we immerse models in, we also are experimenting with immersing the models in different orientations to study the seriation of folding. (3) Agent-based modeling: Models are needed to explore the interactions of subunits in self-assembly experiments from a bottom-up approach. Previous work by Troisi, Wong, and Ratner [39] extol some advantages of applying agent-based modeling; “Common features of the agent rules are as follows. (i) Nonlinearity: The rules have thresholds and if-then conditions. (ii) Locality: Some of the decisions are taken considering only the local environment and not the global average. (iii) Adaptation: The rules may change in time through processes of learning”. Theoretically, we also are “examining whether it is interesting to use knowledge on “morphogenesis” (cf. Alan Turing) and other conceptual ideas of spontaneous self-organization to achieve a desired shape [40].

A classification of self-assembly developed by Mastrangeli, Mermoud and Martinoli [41] lays out a wider spectrum of potential approaches to investigating self-assembly. They note: “SA represents the main embodiment of the bottom-up approach to the fabrication of heterogeneous and articulated micro- and nanosystems. Rooted in, and constantly inspired by, biology and supermolecular chemistry, such an approach is complementary to the top-down fabrication approach established at (though not exclusive to) the macroscale because of its highly de-centralized, massively parallel, and largely unsupervised control, which, together with intrinsic redundancy, makes it also highly robust and, in principle, scalable to the control of larger structures”. We believe that our work that has drawn upon the power of biological self-assembly and self-folding to explore their three fundamental features of biomimetic design: “highly decentralized, massively parallel, and largely unsupervised control”.

## Figures and Tables

**Figure 1 biomimetics-08-00012-f001:**
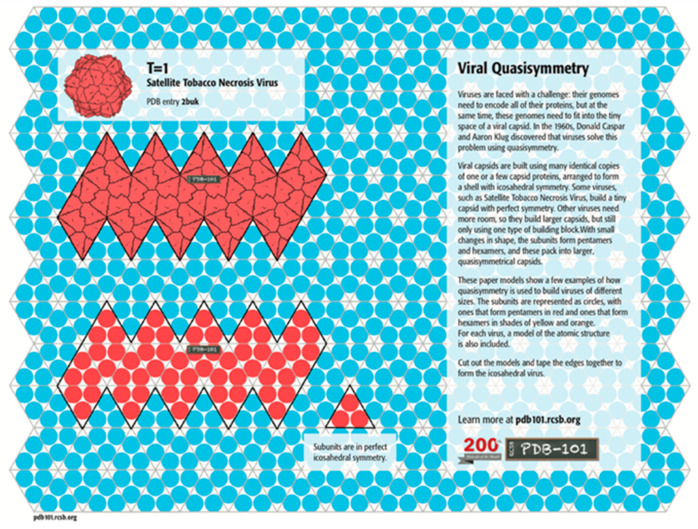
Dürer net of a Satellite Tobacco Necrosis Virus which has sixty capsomere protein subunits (3 per equilateral triangle). Source: CC By 4.0 license PDB-101 https://pdb101.rcsb.org/learn/paper-models/quasisymmetry-in-icosahedral-viruses (accessed on 1 December 2022).

**Figure 2 biomimetics-08-00012-f002:**
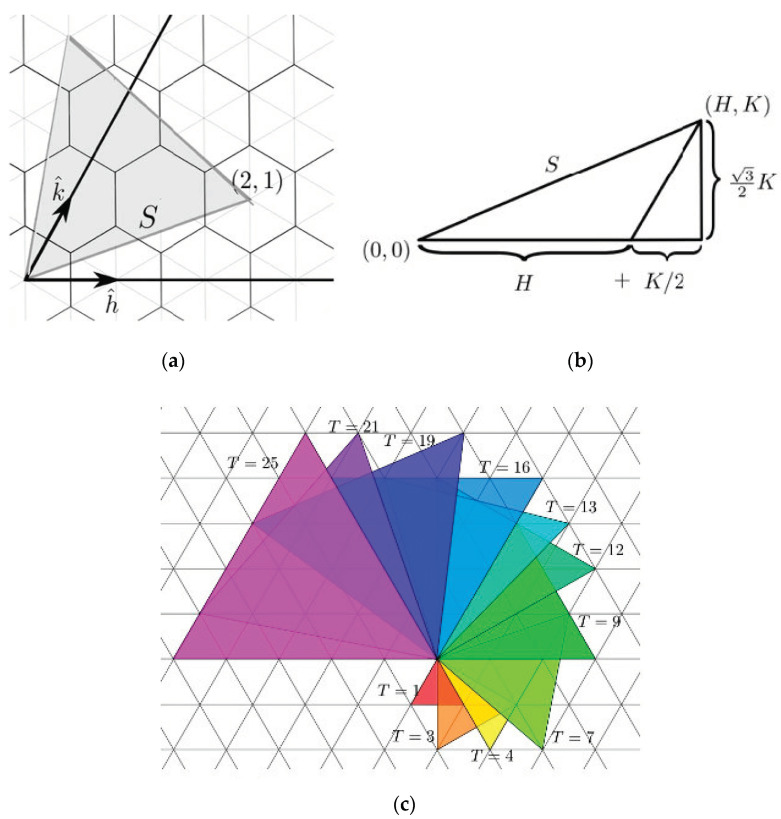
(**a**) Caspar and Klug [2] developed a classification of polyhedral viral capsids based upon a “Triangulation number” where a T-number is mathematically defined as the squared length of each triangle edge: T = (h^2^ + hk + k^2^). It counts the number of symmetrically distinct but quasi-equivalent triangular facets in the triangulation per face of an icosahedron. (**b**) A T1 virus has 60 capsomeres whereas a T3 virus has 180 capsomeres. Note that the volume of a viral capsid increases dramatically as the T number increases. (**c**) Casper and Klug proved allowable triangulations consist of equilateral triangles whose vertices lie on a triangular grid. Source: CC By 4.0 (https://math.stackexchange.com/users/104041/shaun), URL: https://math.stackexchange.com/q/360-3852 (accessed on 1 December 2022).

**Figure 3 biomimetics-08-00012-f003:**
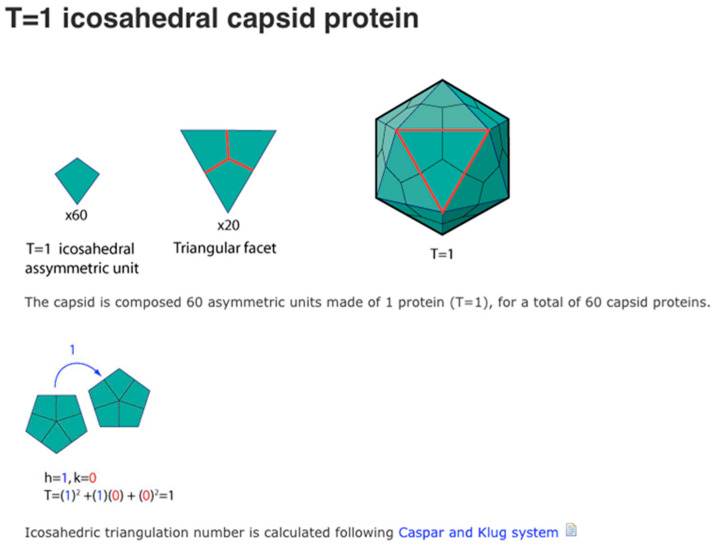
T1 viral capsids have 20 triangular capsomeres made of three rhombic structured protein subunits. Source: Viral Zone, https://viralzone.expasy.org/1057, Swiss Institute of Bioinformatics (accessed on 1 December 2022).

**Figure 4 biomimetics-08-00012-f004:**
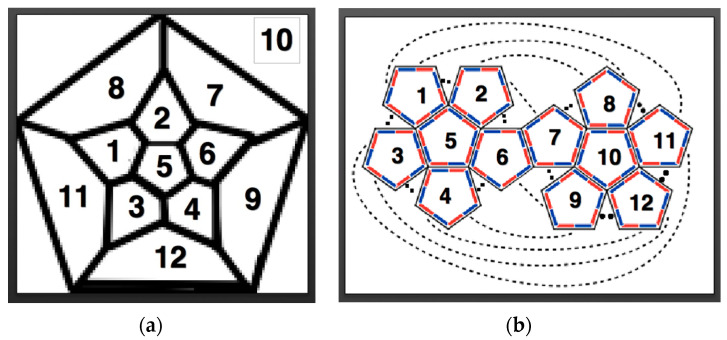
Two different representations of which pentagons in a planar representation of a dodecahedron bind together: (**a**) a Schlegel diagram; and, (**b**) the corresponding Dürer net of a dodecahedron have been numbered 1–12, such that each pentagon with the same number on the corresponding configurations are the same.

**Figure 5 biomimetics-08-00012-f005:**
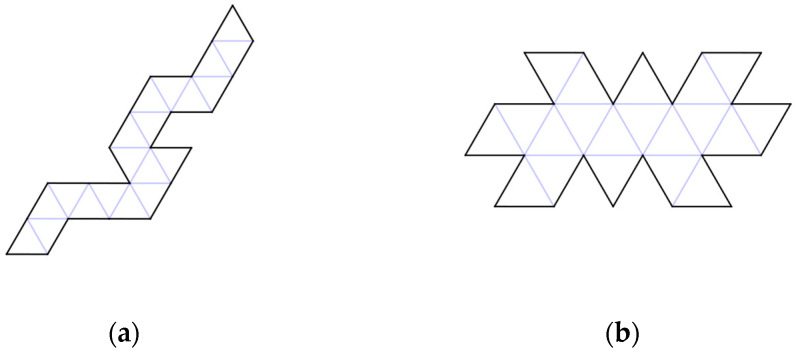
Of the 43,380 configurations of Dürer nets, we challenge the reader to time themselves in folding five different Dürer nets (**a**,**b**) (1, 5a, 5b, 15, 16d). Consider: When do previous gluings make it difficult to make subsequent gluings? How difficult are the last few joinings to enclose the icosahedra?

**Figure 6 biomimetics-08-00012-f006:**
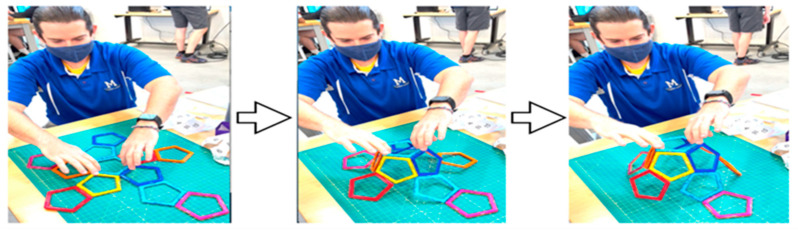
Radial Serial Folding: Middletown, DE high school teacher Matt Juck demonstrated how one Durer net planar configuration of a dodecahedron made with Magformers^TM^ [3] easily folded into a 3D polyhedron.

**Figure 7 biomimetics-08-00012-f007:**
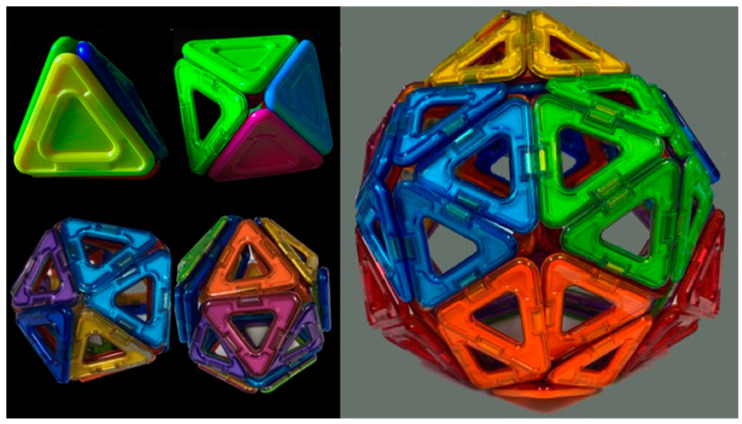
Equilateral triangular Magformers^TM^ [19] units can be assembled to form tetrahedra, octahedra, decahedra, icosahedra and a T1-like 60-piece icosahedra.

**Figure 8 biomimetics-08-00012-f008:**
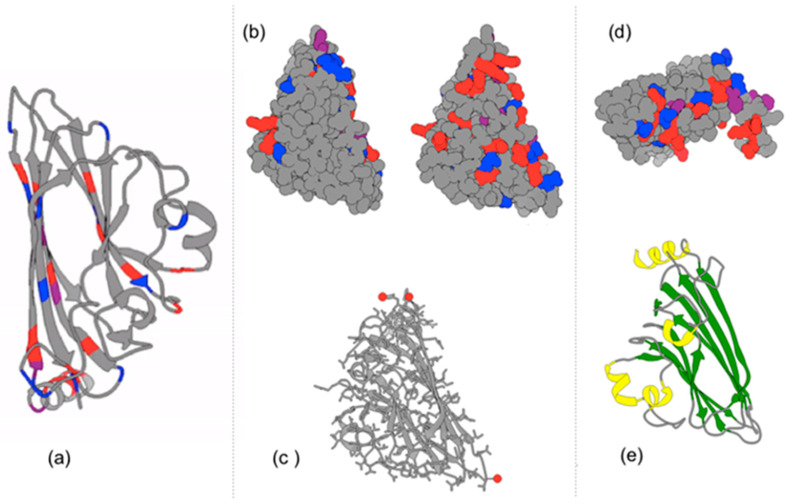
The T1 Satellite Tobacco Necrosis Virus capsomeres are illustrated with Chimera software [insert # reference] using the 3D X-ray crystallography data from the PDB: (**a**) Ribbon structure of beta-pleated sheets and alpha-helices; Positively charges residues are colored red. Negatively charged residues are colored blue. (**b**) Charged residues are highlighted on two different 3D rotations of a capsomere; The front (left) and back (right) of the monomeric subunit. Note the abundance of charged amino acids on the inside of the capsomere, as well as on the sides where capsomeres contact one another. (**c**) Calcium ions are associated with two of the three vertices of the triangular capsid. Calcium ions interact with Asp, Glu, and Thr residues. These are more visible when examining multiple subunits. (**d**) Interestingly, positively charged alpha helices of the capsomeres extend into the interior of the polyhedral capsid which can bind with the negatively charged phosphates in the nucleic acid (RNA) genome in the interior of the capsid; (**e**) Ribbon secondary structure of a capsomere where an alpha-helix (yellow) extends orthogonally with respect to the primary axis of the capsomere’s surface. Note that two large Beta-pleated sheets (green) on the right are antiparallel to one another.

**Figure 9 biomimetics-08-00012-f009:**
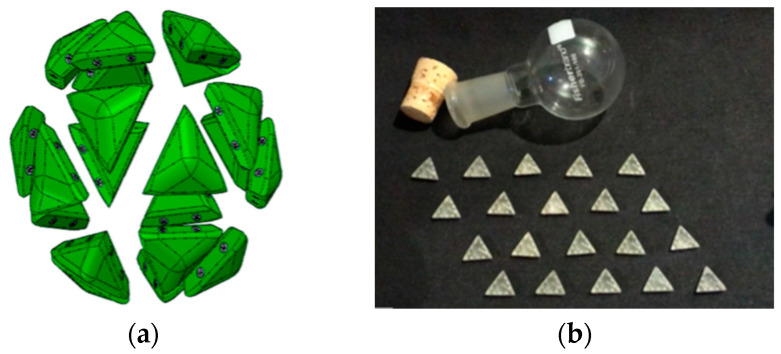
Four successive images from our design of the twenty 3D printed subunits to the self-assembly of the icosahedron in a spherical flask. (**a**) The 20 individual subunits. (**b**) The 20 3D printed subunits laid out on a table before putting into a vessel. (**c**) We found that a spherical vessel (Cook et al. [27]) was more conducive to (**d**) assembling polyhedra than in a cylindrical or an Erlynmeyer flask. See Appendix A and Table 3.

**Figure 10 biomimetics-08-00012-f010:**
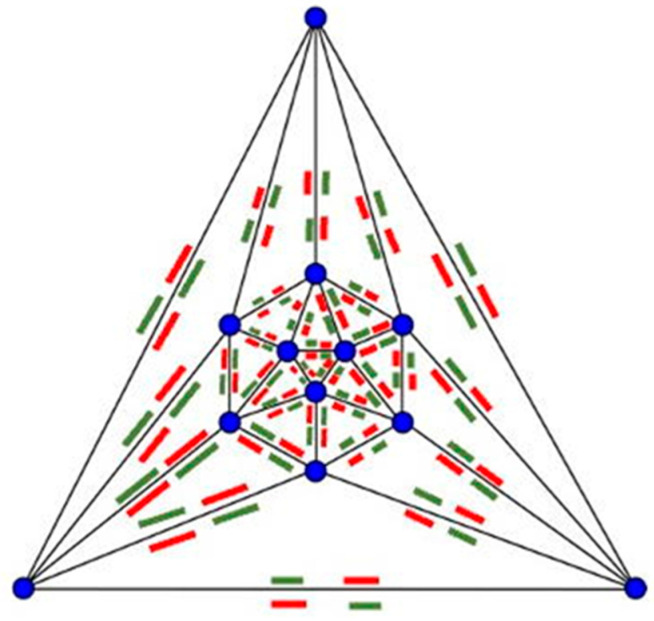
Magnet map on a Schlegel diagram of an icosahedron.

**Figure 11 biomimetics-08-00012-f011:**
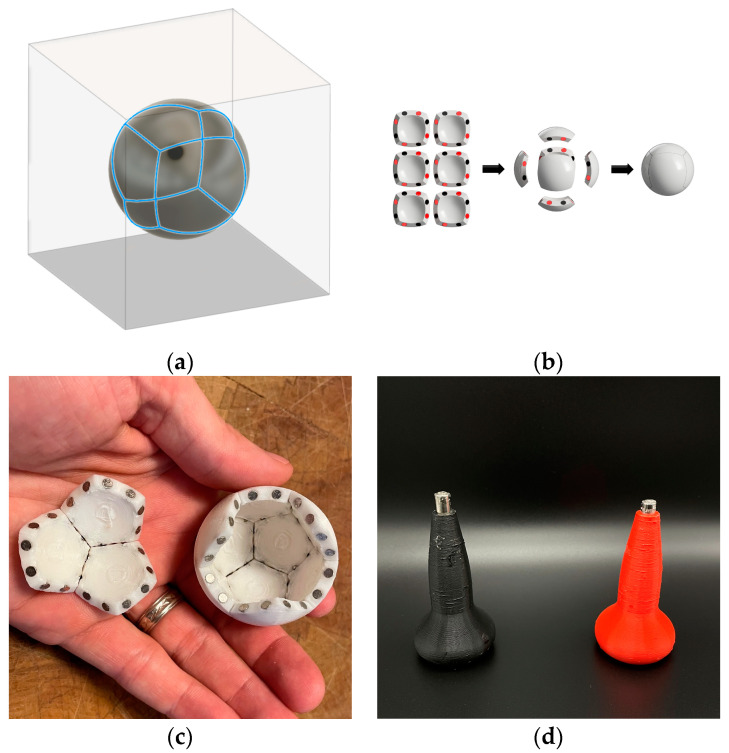
Design of self-assembling pieces of polyhedral. (**a**) The slice lines of the six faces of a cube with a projection onto sphere. (**b**) The six 3D printed subunits with their magnets alternating around each edge of the faces and their assembly back into a sphere. (**c**) Illustration of three and nine pentagon subunits of a partial dodecahedron viewed from the interior. (**d**) North and south oriented magnet holders for correctly inserting magnets into holes on the edges of subunits in a pattern dictated by the Schlegel diagram magnet mapping (see Figure 8).

**Figure 12 biomimetics-08-00012-f012:**
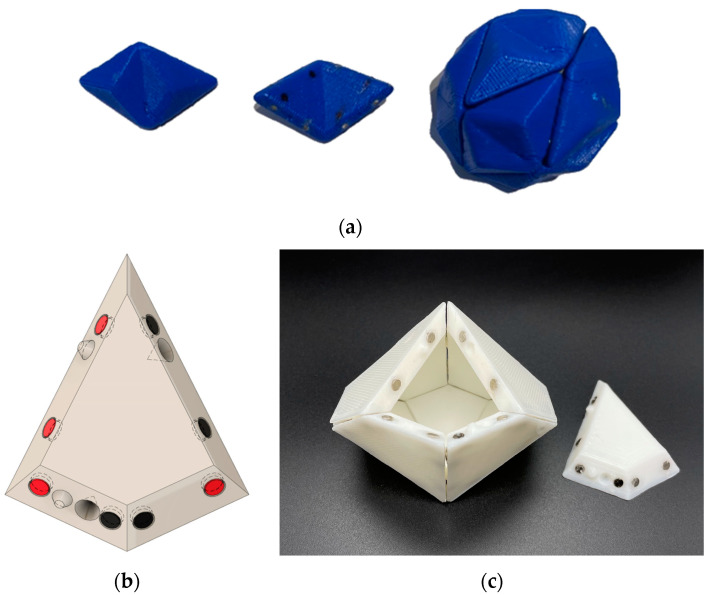
Ten rhombic subunits can self-assemble into a decahedron model of an icosahedral viral capsid. (**a**) A self-assembling decahedron constructed by keeping ten pairs of the subunits in Figure 8 conjoined; (**b**) a rhombic decahedron subunit with its magnet map, and hole and peg arrangement, for 3D printing, image created in Fusion 360; (**c**) a view of an incompletely assembled decahedron; The knobs and holes are to force the long edges to assemble in only one direction. Without them it would be possible for the long edge with black magnets to be attracted to the long edge with red magnets in an “upside-down” configuration. These magnet maps are more complicated than the platonic solid ones and this is where we spent a significant amount of time planning for the T1 model. Because of this model (**b**,**c**) assembles faster and with a higher frequency of completion than model (**a**). We are quite sure that if we used a spherical decahedron the assembly time would improve (**b**,**c**).

**Figure 13 biomimetics-08-00012-f013:**
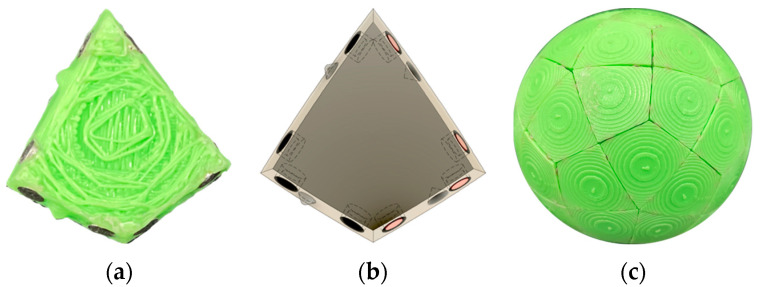
A T1 icosahedral model with sixty identical rhombic subunits. (**a**) An individual 3D printed subunit with its magnets glued into its edges; (**b**) A render of an individual subunit with its magnet map, and hole and peg arrangement, for 3D printing, created in Fusion 360; (**c**) A self-assembled spherical model with 60 subunits.

**Figure 14 biomimetics-08-00012-f014:**
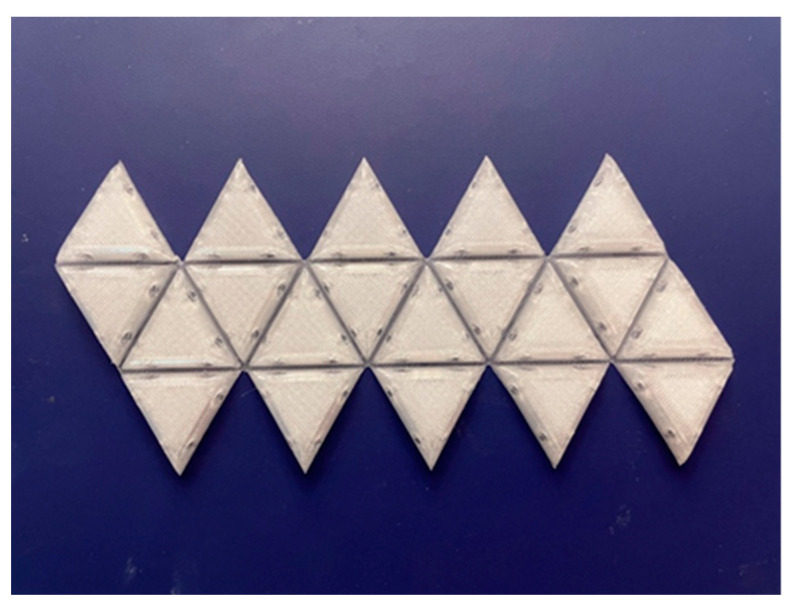
A 3D printed Dürer net of an icosahedron (laid out in a configuration like the origami model of a Satellite Tobacco Necrosis Virus in Figure 1) where the hinges are thinner and are printed in a flat lower layer before the convex domes on each triangular subunit are printed in subsequent layers.

**Figure 15 biomimetics-08-00012-f015:**
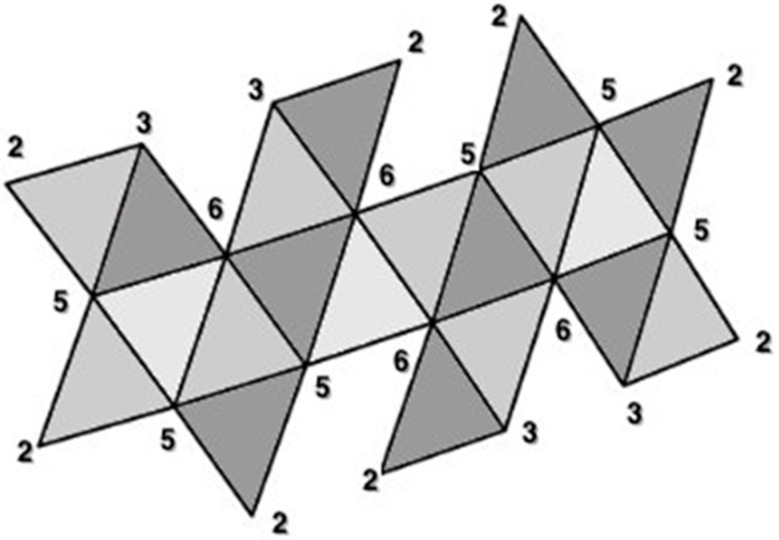
Vertex degrees, given by the number of edges connected to each vertex, are numbered on a Dürer net of an icosahedron. The net contains 20 faces (equilateral triangles; viral capsomeres) with 22 external vertices and 22 external edges. In terms of Euler’s formula V − E + F = 2, this polygon has only 2 faces (all of the colored net is one face and area outside of its boundary is the second face: so 22 − 22 + 2 = 2. Note that once the net is folded, the resulting icosahedron has V = 12, E = 30, and F = 20. This net can be distinguished from the other 43,279 cuts of an icosahedron to generate a planar configuration by a unique Hamiltonian circuit of the degrees of the vertices: beginning with the top left vertex (2-3-6-3-2-6-5-2-5-2-5-2-3-6-3-2-6-5-2-5-2-5-). It is important to distinguish this Hamiltonian circuit from a Hamiltonian path that attaches along one edges of each of the capsomeres [28]. Two additional topological invariants of these structures are the number of vertex connections (see Figure 13) and the number of leaves (see Figure 14). Vertices of degree 6 we will refer to as vertex connections—the edges attached to such vertices are easily joined to one another when the 2D Dürer net if folded into the 3D icosahedron. Vertices of degree 6 on a Dürer net of an icosahedron are called vertex connections. The above configuration thus has 4 vertex connections. Leaves are identified as those polygonal subunits which extend singly (that is, they only share an edge with one adjacent polygonal subunit). Each triangular face with a degree 2 vertex is a leaf. The above configuration thus has 8 leaves.

**Figure 16 biomimetics-08-00012-f016:**
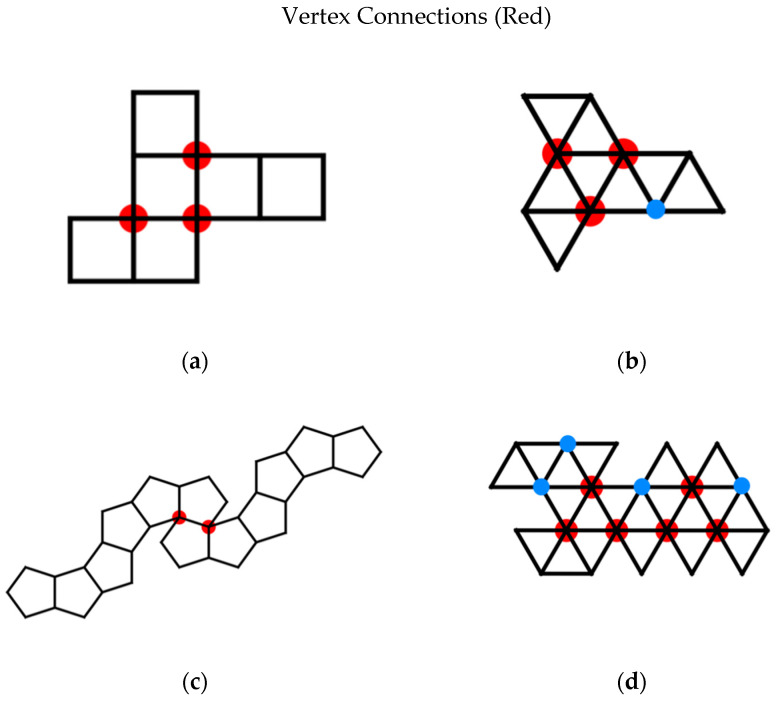
Vertex connections under two different definitions: We prefer the restrictive definition of a vertex connection, shown in red. The traditional definition includes vertices shown in red and blue: (**a**) vertices of degree 4 for Dürer nets of cubes, (**b**) vertices of degree 5 for Dürer nets of octahedra, (**c**) vertices of degree 4 for Dürer nets of dodecahedra, and (**d**) vertices of degree 6 for Dürer nets of icosahedra. We do not think including vertices of degree 4 for Dürer nets of octahedra or vertices of degree 5 for Dürer nets of icosahedra are informative for folding potential.

**Figure 17 biomimetics-08-00012-f017:**
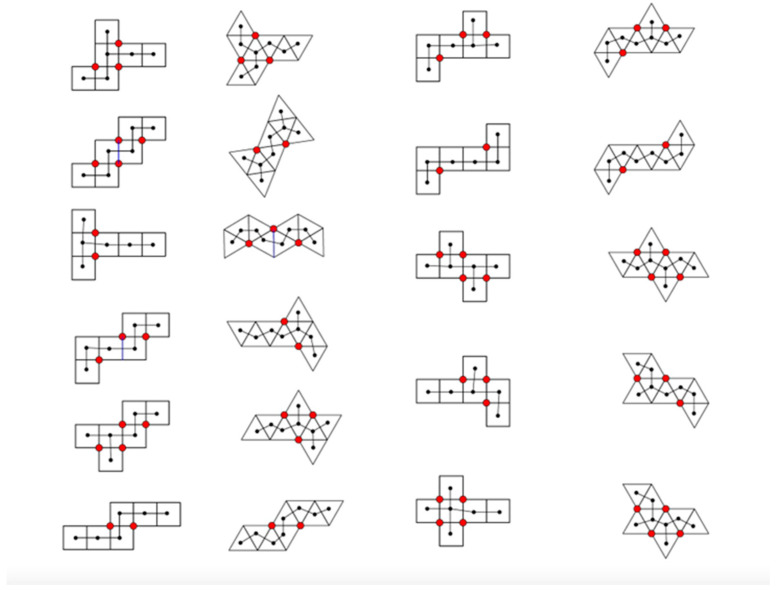
We have redrawn the 11 Dürer nets of cubes and octahedra studied by Dodd, Damasceno, and Glotzer [29] to highlight the position (red) and number vertex connections. We also have added a spanning tree of the centroids of each face.

**Figure 18 biomimetics-08-00012-f018:**
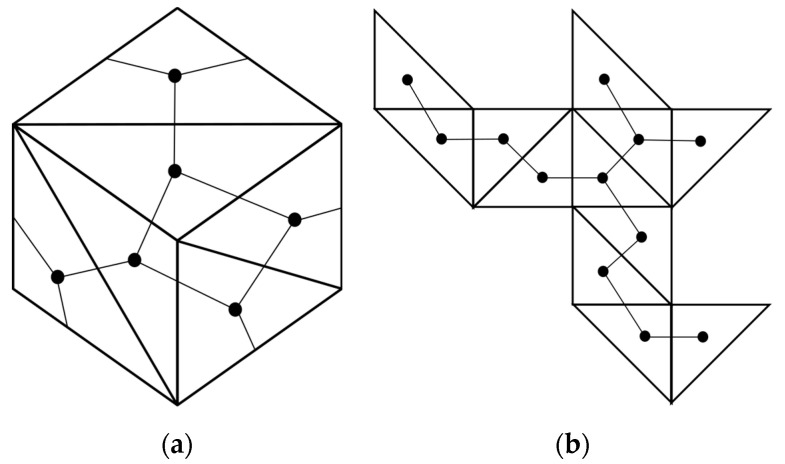
(**a**) Each face of a cube is divided into two equilateral triangles to convert the six faces into (**b**) twelve triangular faces. Once a cube is cut along the edges of the triangles to produce a Dürer net, a spanning tree connects the centroid of each triangle (shown as a black dot on each face), according to their adjacency on the surface of the original cube.

**Figure 19 biomimetics-08-00012-f019:**
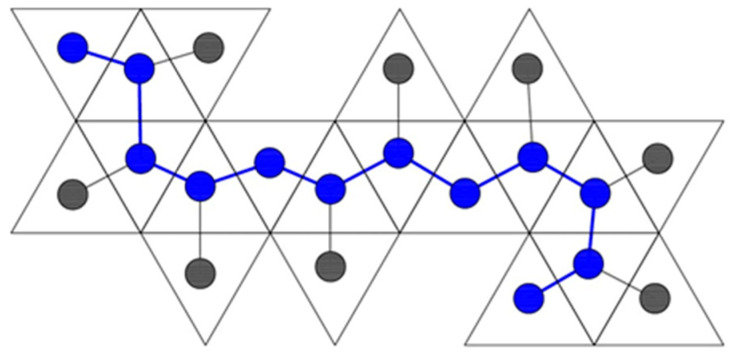
A skeleton graph with its backbone drawn overtop the spanning tree in blue.

**Figure 20 biomimetics-08-00012-f020:**
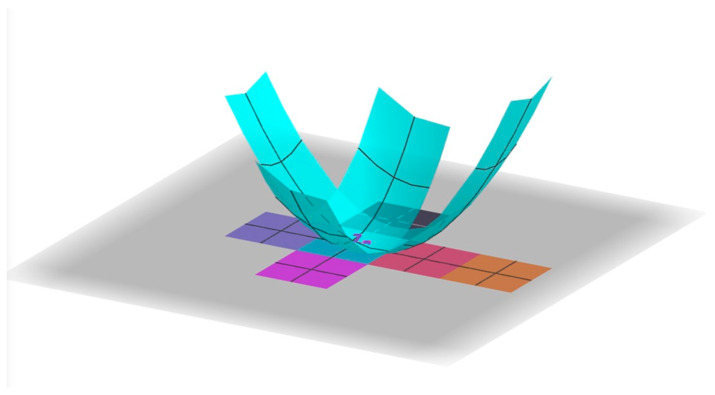
To determine the radius of gyration, we view the net as being embedded in the Cartesian xy-plane, where (x, y) is the center of mass of the net Ω and an integral calculates the volume under a paraboloid with the domain on the xy-plane restricted to the net with the center of mass of the net centered at the origin. Thus, the longer the net, and the less compact it is around its center of mass, the larger quadratic penalty is applied by the integral.

**Figure 21 biomimetics-08-00012-f021:**
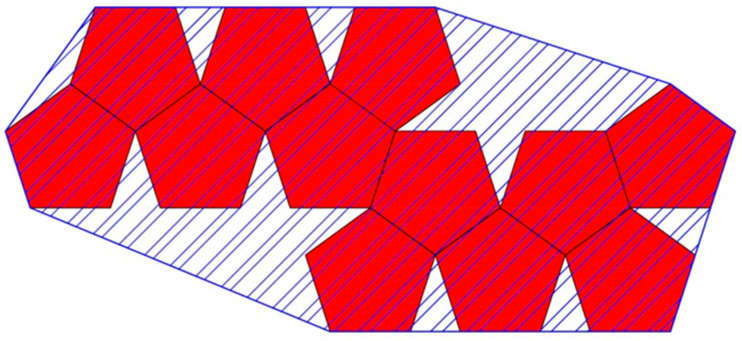
The convex hull (blue line) is defined by a convex polygon attaching all vertices on the exterior of a Dürer net configuration (red polygons) excluding any edges on the periphery that do not contribute to its convexity. The perimeter of the convex hull is the length of the blue line. The area of the convex hull is the area in the interior defined by the blue diagonal lines.

**Figure 22 biomimetics-08-00012-f022:**
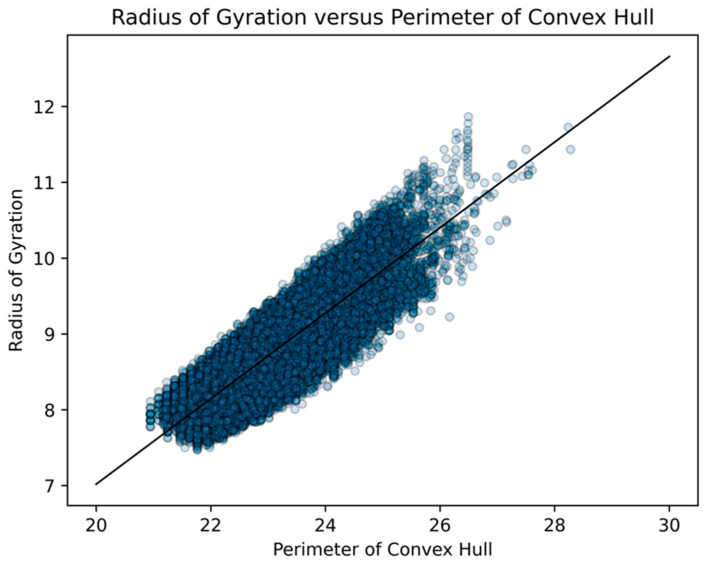
For all 43,380 Dürer nets of dodecahedra, we did a scatterplot of the perimeters of their respective convex hulls with their respective radii of gyration. We standardized the length of an edge of a pentagonal face of a dodecahedron at “1”. The R2 value was 0.738 and the probability of being due to randomness was less than 10^−6^. This significant linear relationship held for all nets with vertex connections ranging from 2 to 10.

**Figure 23 biomimetics-08-00012-f023:**
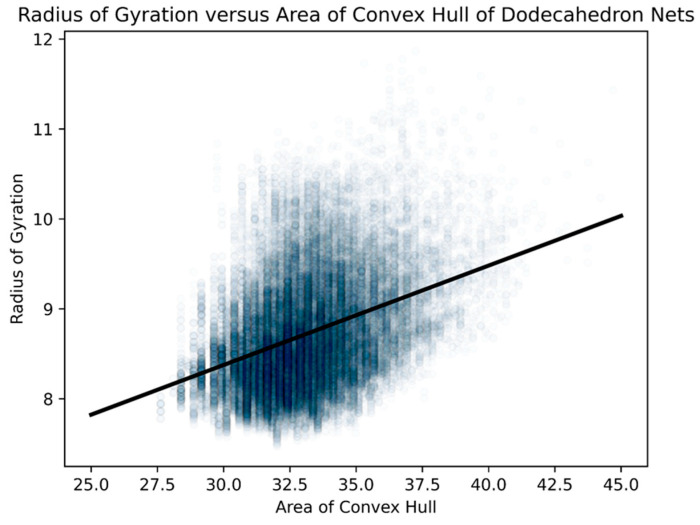
For all 43,380 Dürer nets of dodecahedra, we did a scatterplot of the areas of their respective convex hulls with their respective radii of gyration. We standardized the length of an edge of a pentagonal face of a dodecahedron at “1”. The R2 value was 0.142 and the probability of being due to randomness was less than 10^−6^.

**Figure 24 biomimetics-08-00012-f024:**
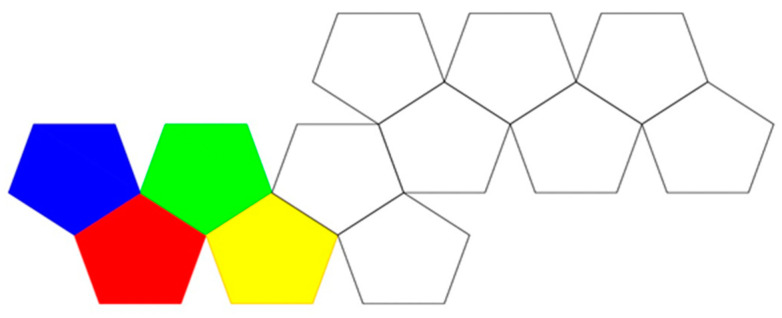
This Dürer is colored to illustrate the order of difficulty of folding at an edge between two individual polygons within the net. The left most blue panel will lift and fold to the inner green panel to its right and the bottom left red panel will lift and fold to the inner yellow panel to its right simultaneously in a true parallel self-folding.

**Figure 25 biomimetics-08-00012-f025:**
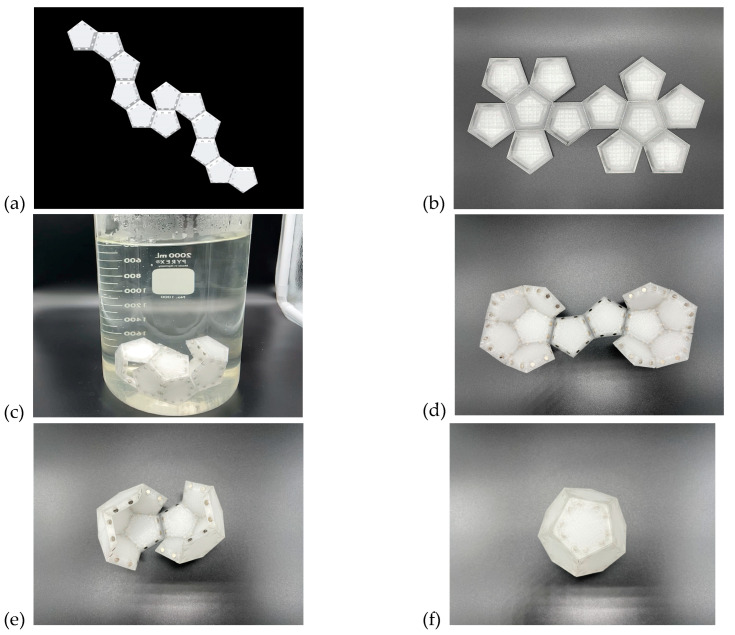
Two 3D-printed Dürer net of dodecahedra with flexible hinges for self-folding were chosen with 2-vertex connections (**a**) and 10-vertex connections (**b**), respectively. In (**c**) the Dürer net of a dodecahedron with 2-vertex connections did not fold successfully even when submerged in a warm water bath for an extended period of time. Appendix A. On the other hand, consistent with our experience with “Radial Serial Folding” (Figure 6), the figures above (**d**–**f**) represent a correct assembly pathway and the one we observe when submerging the Dürer net of a dodecahedron with 10-vertex connections into a warm water bath. In the first figure (**d**) you can see that each of the leaves have folded first forming two almost complete halves of the dodecahedron. These fold first as there is less force required to lift a single leaf into position as compared to the other hinges. To complete the formation of the dodecahedron there are three edges that remain to be closed. If the two on the sides fold before the hinge in the middle the dodecahedron will correctly fold into its final shape. Appendix A. In our experiments, these hinges do indeed fold before the middle hinge. An incorrect pathway occurs if the middle hinge closes before the two remaining that are to its left and right. In this case, the faces of the two halves of the dodecahedron would need to pass through each other to complete the folding procedure.

**Table 1 biomimetics-08-00012-t001:** Classification of assembly and folding approaches to constructing polyhedra.

Type of Organization	Pathway	Guided	Methods of Study	Biological Analogues
True Parallel Self-Folding and Self-Assembly	Parallel	No	Plastic, Nickel and Solder	Viral capsids, ribosomes
Serial Folding and Assembly	Serial	Yes	Origami	Viral capsids, proteins
Template-Assisted (Guided) Serial Folding	Serial	Yes	Origami; tethered plastic	RNA-tethered capsid assembly
Serial Self-Folding	Serial	No	3D printed models in an oriented system	Protein Folding
Radial Serial Folding	Radial	No	Magformers	not yet known
Random Folding	Random	No	Turbulent Systems	in vivo cell conditions

**Table 2 biomimetics-08-00012-t002:** Combinatorial Explosion of the number of Dürer nets for various polyhedra.

Combinatorial Explosion		
Polyhedron	Number of Faces	Number of Dürer Nets
Tetrahedron	4	2
Cube	6	11
Octahedron	8	11
Dodecahedron	12	43,380
Icosahedron	20	43,380
Viral capsid (T = 1)	60	~10^30^

**Table 3 biomimetics-08-00012-t003:** Self-Assembly Times (in seconds) of Ten Trials with the Icosahedron in Different Containers.

Type	1	2	3	4	5	6	7	8	9	10	Avg
Cylindrical	145	174	268	46	72	166	138	367	325	448	214.9
Spherical	58	67	44	97	84	116	86	16	81	28	67.7

**Table 4 biomimetics-08-00012-t004:** The four satisfactory solutions to the pair of linear equations.

A	B	C	D	Nets of This Type
2	4	0	0	4
3	2	1	0	5
4	0	2	0	1
4	1	0	1	1

**Table 5 biomimetics-08-00012-t005:** The number of possible degree distributions for each type of Platonic solid.

Shape	of Faces	of Unique Degree Distributions
Tetrahedron	4	2
Cube	6	4
Octahedron	8	3
Dodecahedron	12	21
Icosahedron	20	9

## Data Availability

Dodd, Damasceno, and Glotzer’s (2018) supplementary datafile of all 86,760 Dürer nets of dodecahedra and icosahedra were invaluable in examining multiple properties of each of these configurations.

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
