# Peer review of "Self-Assembly, Self-Folding, and Origami: Comparative Design Principles"

_biomimetics, 2022, doi:10.3390/biomimetics8010012_

Round 1

Reviewer 1 Report

Attached

Author Response

Thank you for appreciating the merit of our research.

Re (1): Please see the new table 3 where we compare the time of multiple trials for self-assembly of icosahedra in cylindrical and spherical vessels. As we discussed in the article, the geometry of the environment (in this case a container in which the shaking of pieces occur) makes a difference in both the time of assembly and success of assembly.

Re (2) We changed the legend on computing the radius of gyration so that the x,y coordinates are more clearly defined.

Re (3). Unfortunately when we uploaded the original manuscript into the editorial system some page breaks changed and when some multiple images in a single figure were coordinated, some labels were lost. We have re-done a variety of figures to address this problem. We also discovered a number of small errors in figure legends partially due to insertion of new materials and numerous hyphenation problems that occurred in the online re-formatting. We also found that somehow, Table 5 had been lost and it is now re-inserted. We appreciate you pointing out these errors. 

Reviewer 2 Report

In this paper, the authors have used both geometric and topological perspectives to develop a systematic sampling procedure to explore the configurations of the models of a T1 viral capsid with 60 subunits and to test alternatives with 4D printing experiments, and origami models for understanding models of self-assembly of viral capsids. Overall, this paper merits its publication in Biomimetics. The reviewer recommends it to be accepted if the following comments could be properly addressed.

1. In this paper, the authors stated that they found that a spherical vessel was more conducive to assembling polyhedra than cylindrical flasks or Erlynmeyer flasks (Figure 9). Are there any detailed results to support these findings? And could the authors briefly explain the physics behind these findings?

2. In Page 18, the authors used the coordinates x and y to compute the radius of gyration. However, the directions of x and y were not defined clearly, the authors may define them in the figures to make the readers have a better understanding.  

3. In this paper, the label and caption of some figures seem to be not consistent. For example, the Figure 9 has the label of subfigure (d) but there is no caption for this subfigure. Besides, there are no labels of (a) and (b) in Figures 4 and 5. The authors may have a further check to make the figures consistent with their captions in this paper.

Author Response

Dear Reviewer:

Thank you for your very extensive review and interest in our work.

First, we very much agree with your definitions of self-assembly and self-folding. We have inserted a paraphrase of your language up front in the manuscript (see lines 118-121).

Second, Please see the new table 3 where we compare the time of multiple trials for self-assembly of icosahedra in cylindrical and spherical vessels. As we discussed in the article, the geometry of the environment (in this case a container in which the shaking of pieces occur) makes a difference in both the time of assembly and success of assembly.

Third, re your comment about "sweeping classification," we have substantially modified our descriptors in Table 1 to better fit with your concerns and distinctions while keeping categories that respond to the literature we have cited and our own observations. We agree that chaotic was a poor choice given its mathematical definition and have instead focused on the turbulence of the environment in which assembling is occurring. Please especially see the work of Löthman, Per A., Tijmen AG Hageman, Miko C. Elwenspoek, Gijs JM Krijnen, Massimo Mastrangeli, Andreas Manz, and Leon Abelmann. (2020). "A Thermodynamic Description of Turbulence as a Source of Stochastic Kinetic Energy for 3D Self‐Assembly." Advanced materials interfaces 7 (5): 1900963.

Fourth, re origami, we have used it in its traditional sense and have not involved kirigami which we know has also been used in 4D printing studies. 

Fifth, we concur that papers "have been published including compactness, networks, mainpulation of pathways for selecting isomers, degrees of fredom and stable intermediates." We have cited many aspects of that literature as well as attention given to radius of gyration, vertex connections, and leaves. We believe that our focus on adding new features that include spanning trees (both our backbones and skeletons), perimeters and areas of convex hulls, degree distributions, and Hamiltonian cycles of Dürer net vertices, expand both topological and geometric parameters that have either not been considered before or have not been used as design criteria for actual experiments. We do not claim that all of our rules are new, we simply summarize a set of rules that have informed our work.

Sixth, our biomimetic focus is on biological systems. Re the reviewers interest in "self-assembling electronics/computers/furniture in an industrial setting" we feel that is outside the purview of our article. We do note that our colleague Jia-Rey Chang's book: Hypercell: A Bio-inspired Design Framework for Real-time Interactive Architectures (2018) does address some of these issues. 

Reviewer 3 Report

In this article, the authors investigated the differences between self-assembly, self-folding and origami to better design structures that will have bioengineering and biomedical applications. 

I think that this article should introduce a systematic study in the biomimetic design.

For this reason, After long evaluation I recommend this article to be accepted after the fulfill of following “minor revision”:

1)    I believe that it is impossible to find the relative citation in the text. Can you put the number in the text and the relative citation in the section “References”?

2)    You should introduce a section “Experimental” where you explain the steps to fabricate the sub-units obtained by 3Dprinting. The temperature of the warm water bath, the method to calculate the radius of gyration of figure 20, etc….

This will permit to the reader to better understand the fabrication of the subunits and replicate the experiment.

3)    In figure 2 you missed ‘a’ and ‘b’.

4)    In figure 4 you missed ‘a’ and ‘b’.

5)    In figure 5 you missed ‘a’ and ‘b’. 

6)    In figure 9 you missed ‘a’.

7)    In figure 11 you missed ‘a’. 

8)    In figure 12 you missed ‘a’. 

9)    In figure 13 you missed ‘a’.

10)  In figure 22 et 23 can you put the measure units if any?

Author Response

Dear Reviewer,

Thank you for your interest in our work.

Re (1) we have used the author(s)' last name and year of publication to correspond to each of the references in the bibliography.  

Re (2) the details of experimental are associated with the appropriate section. For example, to print a self-folding polyhedron we describe the 3D printing process as occurring in two layers in order to create hinges (see the legend on Figure 14). Re self-assembly, the four main aspects were described:  the magnet map using Schlegel diagrams, convex surface of the units to correspond to the concave surface of the vessel that they are assembled in, the strength of magnets relative to weight of pieces, and the angle of cuts from the 3D polyhedron to begin with. We feel that we have elaborated both of these as general design principles. We have added a paragraph on the inexpensive nature of our materials and that our experiments can easily be reproduced on a home 3D printer. We have modified the legend on calculating the radius of gyration to make it clearer and modlfy correspond with the two different approaches to its calculation. The temperature of the water for folding is a subject of future experiments as we are eager to show the proof of concept of our self-folding models. We plan on multiple experiments where we will examine the temperature, time, degree of completion, effect of turbulence in the environment, etc. We have performed experiments since submitting which accounts for a new table of multiple trials of self-folding of icosahedra at the same temperature in different shaped containers (spheres versus cylinders). As per the reviewer, we are eager that others be able to  "replicate the experiment" as we have done extensive and successful outreach with high school teachers and students using and building our models.

Re 3 through 9, these labels were removed somehow in the conversion online. They have all been added back in. 

Re (10), Figures 22 and 23, we standardized the length of an edge of a pentagonal face of a dodecahedron at "1". This note has been added to the legend of each figure.

 Thank you for your request for clarifications.

Round 2

Reviewer 1 Report

The authors have largely addressed issues. I see that the authors have used images from the internet (stackexchange) and other sources without appropriate acknowledgment in the caption. Typically when an image is reused, the authors need to request permission and explicitly state. Image reprinted from xxx, and cite the copyright holder or else if it is open access state that. 

Author Response

Dear Reviewer,

The only images that we did not create ourselves are 1, 2, 3, and 11:

Figure 1: The source is cited and their images are available via Creative Commons: Here is their license.

https://www.rcsb.org/news/feature/611e8d97ef055f03d1f222c6

PDB adopts a standard Creative Commons open source license

Figure 2: The source is cited and their images are available via Creative Commons: Here is their license.

https://meta.stackexchange.com/questions/333089/stack-exchange-and-stack-overflow-have-moved-to-cc-by-sa-4-0

Stack Exchange network will be available under the terms of version 4.0 of the Creative Commons Attribution-ShareAlike (CC BY-SA) license.

Figure 3: The source is cited and their images are available via Creative Commons: Here is their license.

https://viralzone.expasy.org/search?query=creative+commons

Core content: ... licenses by 4.0 Creative Commons Attribution 4.0 ...
... creativecommons.org licenses by 4.0 Creative Commons License

Figure 18. The source is cited in the legend and I have written the first author for permission. He left George Mason recently for Google. If I don't get a response quickly, the image is easily reproduced as a drawing in a different orientation without violation of copyright as it  is a generic mathematical idea.